# Timing of initiation of anti-retroviral therapy predicts post-treatment control of SIV replication

**Mykola Pinkevych[1], Steffen S. Docken[1], Afam A. Okoye[2], Christine M. Fennessey[3], Gregory Q. Del Prete[3], Maria Pino[4], Justin L. Harper[4], Michael R. Betts[5], Mirko Paiardini[4,6], Brandon F. Keele[3], Miles P. Davenport[1]***

**1** Infection Analytics Program, Kirby Institute for Infection and Immunity, UNSW Australia, Sydney, New South Wales, Australia, **2** Vaccine & Gene Therapy Institute, and Oregon National Primate Research Center, Beaverton, Oregon, United States of America, **3** AIDS and Cancer Virus Program, Frederick National Laboratory for Cancer Research, Frederick, Maryland, United States of America, **4** Division of Microbiology and Immunology, Emory National Primate Research Center, Emory University, Atlanta, Georgia, United States of America, **5** Department of Microbiology and Center for AIDS Research, Perelman School of Medicine, University of Pennsylvania, Philadelphia, Pennsylvania, United States of America, **6** Department of Pathology and Laboratory Medicine, School of Medicine; Emory University, Atlanta, Georgia, United States of America

* m.davenport@unsw.edu.au

**Data Availability Statement:** Data and the analysis used in the current study are available at GitHub repository via the link https://github.com/iap-sydney/Post-treatment_control_of_SIV_

## Abstract

One approach to 'functional cure' of HIV infection is to induce durable control of HIV replication after the interruption of antiretroviral therapy (ART). However, the major factors that determine the viral 'setpoint' level after treatment interruption are not well understood. Here we combine data on ART interruption following SIV infection for 124 total animals from 10 independent studies across 3 institutional cohorts to understand the dynamics and predictors of post-treatment viral control. We find that the timing of treatment initiation is an important determinant of both the peak and early setpoint viral levels after treatment interruption. During the first 3 weeks of infection, every day of delay in treatment initiation is associated with a 0.22 $\log_{10}$ copies/ml decrease in post-rebound peak and setpoint viral levels. However, delay in initiation of ART beyond 3 weeks of infection is associated with higher post-rebound setpoint viral levels. For animals treated beyond 3 weeks post-infection, viral load at ART initiation was the primary predictor of post-rebound setpoint viral levels. Potential alternative predictors of post-rebound setpoint viral loads including cell-associated DNA or RNA, time from treatment interruption to rebound, and pre-interruption CD8+ T cell responses were also examined in the studies where these data were available. This analysis suggests that optimal timing of treatment initiation may be an important determinant of post-treatment control of HIV.

## Author summary

HIV infection can be effectively treated with antiretroviral therapy. However, the virus persists in a latent form despite treatment, and 'rebounds' to high levels if treatment is

replication.git The previously published data: C. M. Fennessey et al., Genetically-barcoded SIV facilitates enumeration of rebound variants and estimation of reactivation rates in nonhuman primates following interruption of suppressive antiretroviral therapy. PLoS Pathog. 13, e1006359 (2017). M. Pinkevych et al., Predictors of SIV recrudescence following antiretroviral treatment interruption. Elife 8, e49022 (2019). A. A. Okoye et al., Early antiretroviral therapy limits SIV reservoir establishment to delay or prevent post-treatment viral rebound. Nat. Med. 24, 1430-1440 (2018). M. Pino et al., Limited impact of fingolimod treatment during the initial weeks of ART in SIV-infected rhesus macaques. Nat. Commun. 13, 5055 (2022). J. Harper et al., CTLA-4 and PD-1 dual blockade induces SIV reactivation without control of rebound after antiretroviral therapy interruption. Nat. Med. 26, 519-528 (2020).

**Funding:** This work was supported by NIH grants NIAID P01AI131338 (to MPD, MP, and MRB); NIAID R21/R33AI116171, NHLBI, NIDDK, NIMH, NINDS, NIDA, NIAID, UM1AI164562, NIAID U42OD011023, and P51OD011132 (to MP). MPD is supported by an NHMRC Investigator grant (1173027) and an NHMRC Program grant (1149990). AAO is supported by NIH grants UM1AI126611, P51OD011092 and AI144993. This project has been funded in whole or in part with federal funds from the National Cancer Institute, National Institutes of Health, under Contract No. 75N91019D00024 and HHSN261201500003I (to BFK). The content of this publication does not necessarily reflect the views or policies of the Department of Health and Human Services, nor does mention of trade names, commercial products, or organizations imply endorsement by the U.S. Government. The funders had no role in study design, data collection and analysis, decision to publish, or preparation of the manuscript.

**Competing interests:** I have read the journal's policy and the authors of this manuscript have the following competing interests: MRB is a paid consultant of Interius Biotherapeutics. All other authors have declared that no competing interests exist.

stopped. A number of approaches are now being investigated that might reduce the level of viral rebound following treatment interruption. However, little is known about how variation in infection history (e.g., timing of treatment initiation) influences rebound and may bias observed effects of trial treatments. Here, we aggregated data across multiple studies of SIV infection (a non-human primate model of HIV) to investigate what factors determined the level of viral rebound after treatment interruption. We found that the time from infection to when antiretroviral therapy was initiated was a major factor influencing average viral levels in the first months post treatment. Over the first three weeks of SIV infection, delaying treatment reduced levels of viral rebound when treatment was stopped. This is assumed to occur because increased exposure to the virus induces a more potent immune response. However, after three weeks, later initiation of antiretroviral therapy was associated with higher SIV levels after stopping treatment. Future work is needed to investigate whether the same patterns of viral rebound are seen in HIV infection.

## Introduction

Antiretroviral therapy (ART) can suppress HIV replication for prolonged periods, but cessation of treatment usually results in a rapid rebound of plasma viral loads in most individuals. However, a small proportion of individuals have been shown to experience good control of HIV plasma viremia following treatment interruption, and this proportion is thought to be greater in patients treated in early infection [1–3]. If persisting low levels of plasma viremia in the absence of therapy could be induced in a high proportion of individuals, this might provide a 'functional cure' of HIV. The frequency of spontaneous post-treatment control varies significantly across different studies and depends in part on the definition of control [1,4–6]. A recent study suggested that around 4% of individuals treated in chronic infection exhibit post-rebound control of HIV in the first 6 months after treatment interruption (defined as 2/3 of viral levels being below 400 copies/ml), and that this proportion rises to 13% in patients treated in early infection [7]. However, studies of subjects treated extremely early in Fiebig 1 [8] failed to show a high rate of post-rebound control [9]. Thus, the relationship between ART treatment timing and post-rebound controls remains poorly understood.

A significant limitation in understanding post-treatment control is identifying a sufficient number of well characterized individuals for comparison. In addition, many treatment interruption studies currently require retreatment of individuals as soon as viral loads reach between 1000 and 100,000 HIV copies/ml, so a true 'setpoint' HIV viral load is rarely documented [10]. Therefore, identifying the determinants of post-treatment control, which may occur only after reaching a rebound viremia peak, and whether such control can be predicted by events during initial infection or early after treatment interruption is a priority.

Non-human primates have been increasingly used to study the dynamics of SIV latency and treatment interruption [11–15]. However, even in this well controlled animal model, the determinants and dynamics of post-rebound control of SIV replication are incompletely understood. Here, we analyse data from 124 macaques in 10 different studies from three different institutions to investigate the determinants of post-rebound control of viral replication (Fig 1). We find that the timing of ART is a major predictor of both the level and duration of post-rebound SIV control.

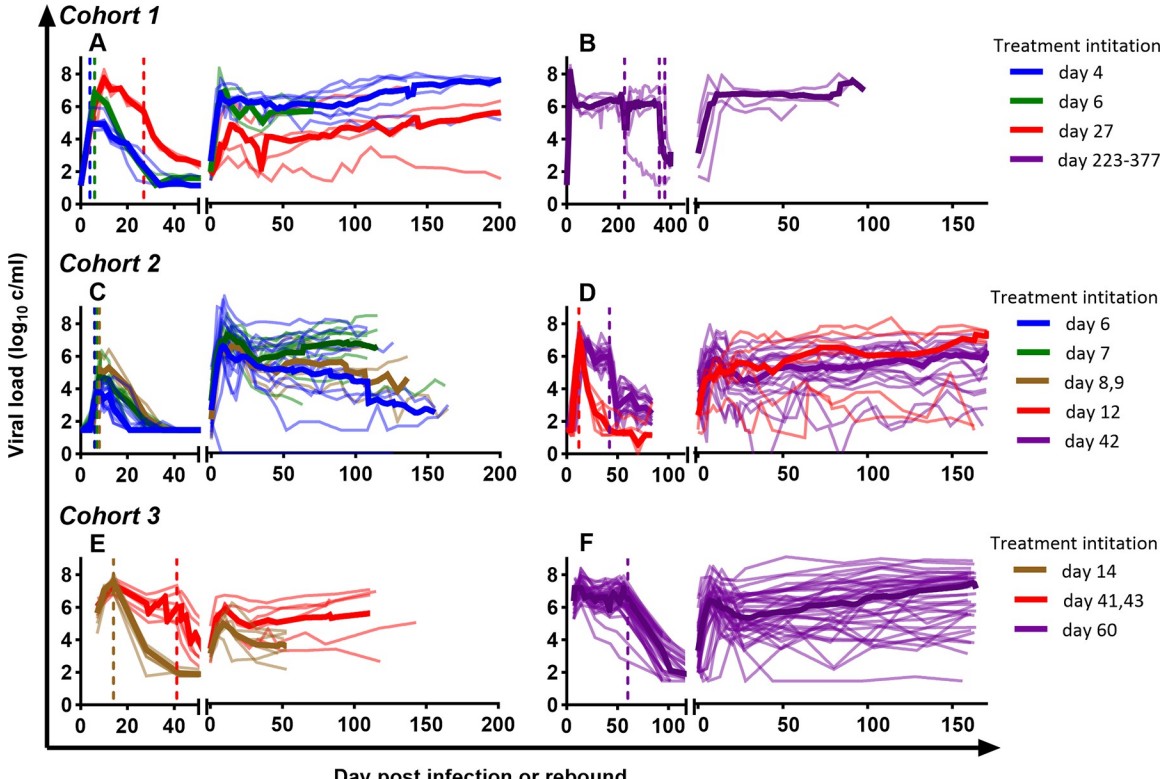

**Fig 1. Dynamics of viral load by cohorts.** Thick coloured lines are the group median viral loads, thin lines of the matched colour are the individual viral loads of monkeys from the same group. Vertical dashed lines indicate the timing of antiretroviral treatment initiation of monkeys in the group with respectively coloured lines.

## Results

### Timing of ART initiation predicts early viral control after treatment interruption

To understand viral dynamics after treatment interruption in SIV infection, we compared set-point viral load in untreated primary infection with the setpoint viral load observed after treatment interruption (Fig 2). We focused on the early setpoint viral level, which we defined as the time-weighted $\log_{10}$ viral load between days 30 and 60 after the first detection of the virus in plasma (Fig 2A) after infection or after rebound (shorter periods were used in some animals due to data availability, detailed in Table A in S1 Text). In our analysis, we used animals from cohorts that were treated with ART between days 4 and 377 post-infection. These animals differed in the initial infection regime, timing and nature of antiretroviral therapy, as well as duration of therapy and frequency of sampling. A brief outline of these studies is presented in Table 1 and full details in Table A in S1 Text.

We first visualised how the timing of initiation of antiretroviral therapy affected the setpoint viral load after treatment interruption (Fig 2B). This analysis shows a clear trend of later initiation of ART being associated with a decline in setpoint viral load, until some timepoint between day 14 and 27. At later timepoints beyond day 27, this trend was reversed, and a delay in treatment was associated with higher mean post-rebound setpoint viral load. We fitted a piecewise regression model to the log10 viral setpoints, assuming a constant decrease in setpoint with later treatment until a nadir timepoint, followed by a later rise in setpoint viral levels

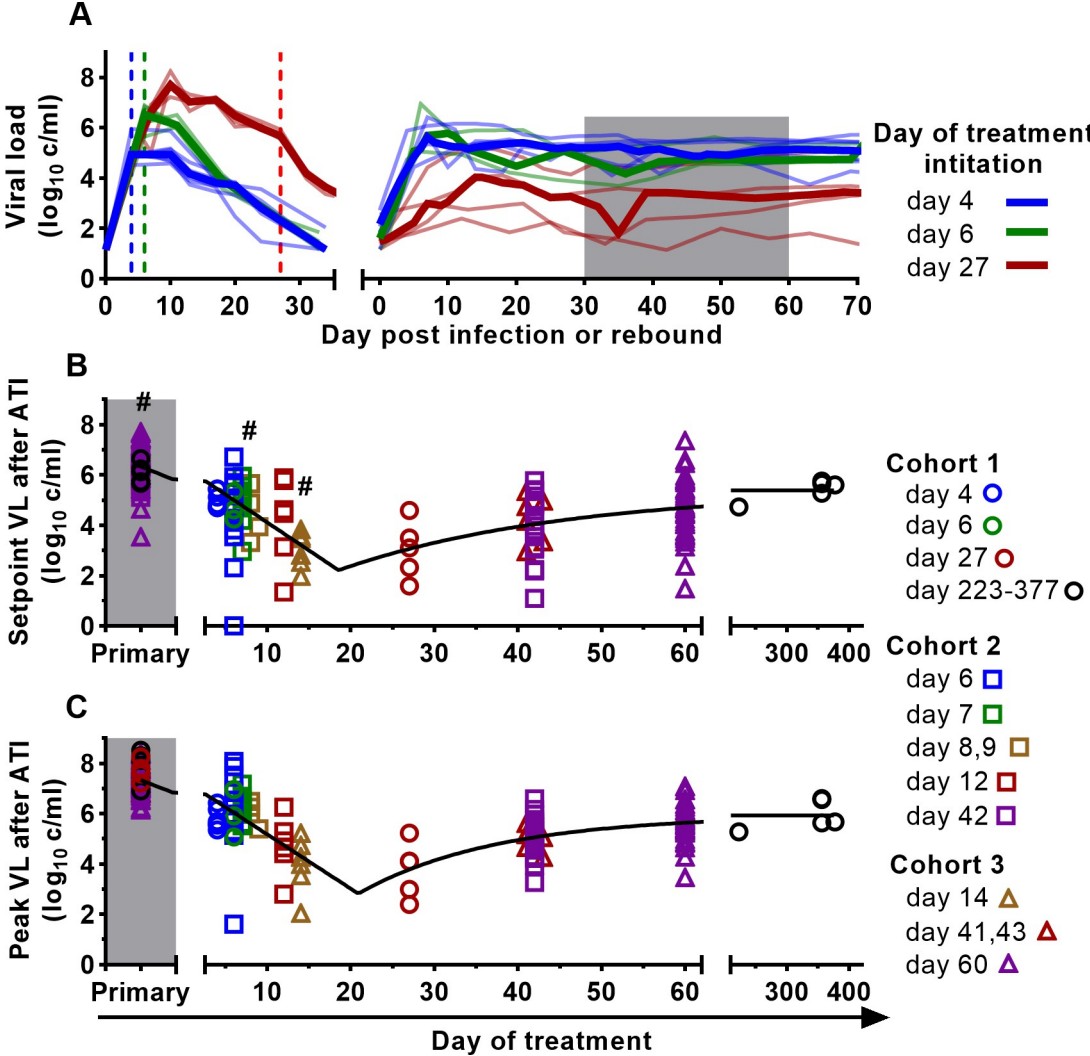

**Fig 2. Timing of treatment affects early post-rebound viral levels. A.** Example of viral load trajectories for animals treated on days 4 (blue), 6 (green), 27 (red) from Cohort 1. Vertical dashed lines indicate the timing of antiretroviral treatment initiation. Shaded area indicates time of early post-treatment setpoint viral load (days 30–60 post-rebound). **B, C.** The relationship between day of anti-retroviral treatment initiation and setpoint viral load (n = 122) (**B**) or peak viral load (n = 124) (**C**) after treatment interruption. Black lines show the best fit of the nonlinear regression (see methods and Formula (2)) that estimates the early decrease, inflection point, and later increase in viral levels. When fitting model (2), we incorporate the data for set point and peak viral load at primary infection (Primary) by defining the timing of treatment to be 0 days for these data points. The best-fit parameters are in the Table B in S1 Text. The SIV inoculum, duration of treatment, and other interventions are shown in Table A in S1 Text. # Time-weighted setpoint viral loads were averaged over shorter time intervals for some animals (see Table A in S1 Text).

with later treatment (described by Formula (2)). We estimated that during early infection, the post-rebound set point viral level is decreased by 0.22 $\log_{10}$ copies/ml (95% CI = 0.18–0.27) for every day for which treatment initiation is delayed. The greatest control is estimated to occur around 18.5 days post-infection (95% CI = 13.6–23.5), after which this control begins to be lost with further delay in treatment. However, although the setpoint viral level is predicted to increase with later treatment to a maximum of 5.4 $\log_{10}$ copies / ml (95% CI = 4.6–6.2), this is below the mean setpoint viral levels seen in an untreated primary infection (6.3 $\log_{10}$ copies/ ml, 95% CI = 6.1–6.6) (see Table B in S1 Text for the best-fit parameters). This finding is

**Table 1. A short summary of the experimental studies used in the current analysis.** The first column is the identifier as it appears in the text. The more detailed parameters of the studies are presented in the Table A in S1 Text.

| Study identifier | Treatment schedule | | N | Reference |
|---|---|---|---|---|
| | Initiation (days post-infection) | Duration (days) | | |
| National Cancer Institute (Cohort 1) | | | | |
| 1.a | 4 | 301–478 | 6 | [16] |
| 1.b | 6 | 81 | 3 | [16] |
| 1.c | 27 | 322–476 | 5 | [17] |
| 1.d | 223–377 | 218–410 | 5 | |
| Oregon Regional Primate Center (Cohort 2) | | | | |
| 2.a | 6–9 | 602–605 | 27 | [11] |
| 2.b | 12 | 356 | 12 | [11] |
| 2.c | 42 | 928 | 18 | [11] |
| Emory National Primate Research Center (Cohort 3) | | | | |
| 3.a | 14 | 205/206 | 7 | [18] |
| 3.b | 41,43 | 357 | 6 | [18] |
| 3.c | 60 | 265–438 | 41 | [15] |

consistent with increased exposure to the virus early in the first few weeks of primary infection leading to increased immunity and reduced early setpoint viral levels post-ATI. However, if treatment is delayed beyond a certain point (around day 18 post-infection), ongoing exposure to infection is associated with an increase in setpoint viral level after treatment interruption, suggesting a role for factors such as immunosuppression, immune exhaustion or viral escape. The descriptive statistics and the comparisons of the setpoint viral load by institutional cohorts are in Fig A in S1 Text.

## Timing of ART initiation also affects post-rebound viral peak

The observed changes in post-rebound viral setpoint with treatment timing suggest effects of immune control on viral replication. To investigate this further, we also analysed whether treatment timing affected peak viral levels after treatment interruption (Fig 2C). Visual inspection suggested that later treatment was associated with reduced post-rebound peak viral levels out to a nadir timepoint, followed by a gradual increase in peak post-rebound viral levels with later treatment. We fitted the same model (see Formula (2)) to determine the relationship between treatment timing and the post-rebound peak of virus. We estimated that post-rebound peak viral levels decrease by 0.22 $\log_{10}$ copies/ml for every day for which treatment initiation is delayed in early infection. The nadir of the post-rebound peak viral levels occurs when treatment was initiated at around 20.9 days post infection (95%CI = 17.1–24.7) (see Table B in S1 Text for the best-fit parameters). To test whether the trajectories of peak and setpoint viral levels followed similar patterns, we also fitted both datasets simultaneously, allowing only the initial setpoint and peak viral levels to be different. This provided a good fit to the data, suggesting that overall shape of the drop (and subsequent rise) of viral levels with later treatment initiation is very similar between setpoint and peak post-rebound viral levels (p = 0.91, F-test; See Table B in S1 Text for the best fit parameters and Table C in S1 Text for goodness of fit of models with various parameters differing between datasets). However, peak viral level after treatment interruption was consistently a mean of 0.98 $\log_{10}$ copies/ml higher than rebound setpoint viral levels, regardless of the timing of treatment initiation (Fig B in S1 Text). This suggests that similar factors may determine both the post-rebound peak and

setpoint viral levels, but with the setpoint being consistently around 1 $\log_{10}$ copies/ml lower than the peak (Fig D.A-C in S1 Text). Interestingly, the drivers of rebound peak and early setpoint are not the same as those of viral growth rate, as the viral growth rate during early rebound was not strongly predictive of the later setpoint viral load (Fig D.D-F in S1 Text). Our combined analysis of post-rebound peak and setpoint viral levels suggest that the optimal control of both post-rebound peak and setpoint viral levels is achieved after treatment initiation around days 18–21 days post-infection. For clarity in later discussion we refer to the 'optimal time of treatment initiation' as being around day 20 post-infection.

## Factors in primary infection predicting post-rebound viral control

The analysis above suggests that delays in initiation of ART may contribute to the induction and subsequent loss of immunity in primary infection, which later contributes to post-ATI viral control. A likely explanation is simply that treatment timing may lead to differences in exposure to virus pre-treatment, which affects subsequent induction and loss of immunity. However, there was also significant variation in the plasma virus levels for animals in the same cohort treated on the same day post-infection, suggesting differences in viral replication capacity between animals. For example, macaques treated on day 12 post-infection (Cohort 2.b) had viral loads at treatment ranging from 5.8–7.7 $\log_{10}$ copies ml$^{-1}$. Therefore, if differences in viral level alone predict post-treatment viral control, then we should be able to see correlations with pre-treatment viral level independent of treatment timing. To understand whether the kinetics of virus replication and levels of plasma virus in primary infection are predictive of post-treatment control, we analyzed whether either viral load at treatment initiation or at peak viral load predicted the later post-rebound setpoint viral levels (Fig 3A and 3B). We found that, when we considered all cohorts together, viral load at treatment initiation and peak viral load in primary infection both correlated poorly with post-rebound setpoint viral load (linear regression adjusted $R^2$ = 0.040, p = 0.016 and adjusted $R^2$ = -0.008, p = 0.79 respectively. These correlation analyses, as well as those of other parameters of interest from the Results section, are summarised in the Table H in S1 Text).

However, since early and late treatment had different effects on post-rebound setpoint viral levels (Fig 2), we also split the cohorts depending on time-of-treatment and separately analyzed animals treated before or after day 20 (the inflection point predicted by the modeling). For animals treated prior to day 20, viral load at treatment remained a poor predictor of post-rebound setpoint viral levels (Fig 3C. $R^2$ = 0.0016, p = 0.31). However, consistent with the results in Fig 2, timing of treatment was significantly associated with post-rebound setpoint viral levels in the animals treated before day 20 (Fig 3D, p = 0.0077). However, because of the large variation in individual rebound viral loads amongst animals treated on a given day, timing of treatment only explained a small amount of the observed individual viral load variation (adjusted $R^2$ = 0.13). We also analyzed whether, for animals treated on the same day, having an increased viral level at treatment was associated with increased post-rebound setpoint viral levels (i.e., a nested model where we tested whether a model of time-of-treatment was improved by adding data on viral load at treatment. Fig C.A in S1 Text). However, adding information on viral load at treatment to the model did not significantly improve the fit (adjusted $R^2$ = 0.15, p = 0.16, F-test, comparing day of treatment only vs. day of treatment plus viral level model, Fig C.A in S1 Text). Given that animals treated on the same day could vary >10-fold in the plasma viremia, it is perhaps surprising that plasma viral loads were not more predictive of post-rebound viral kinetics. Nonetheless, this analysis demonstrates that timing of treatment initiation is the major predictor of post-rebound setpoint viral levels for animals treated prior to optimal time of treatment initiation.

We also analysed the predictors of post-rebound setpoint viral load for animals treated later in primary infection (treatment initiated ≥ 20 days post-infection). In these animals, viral load

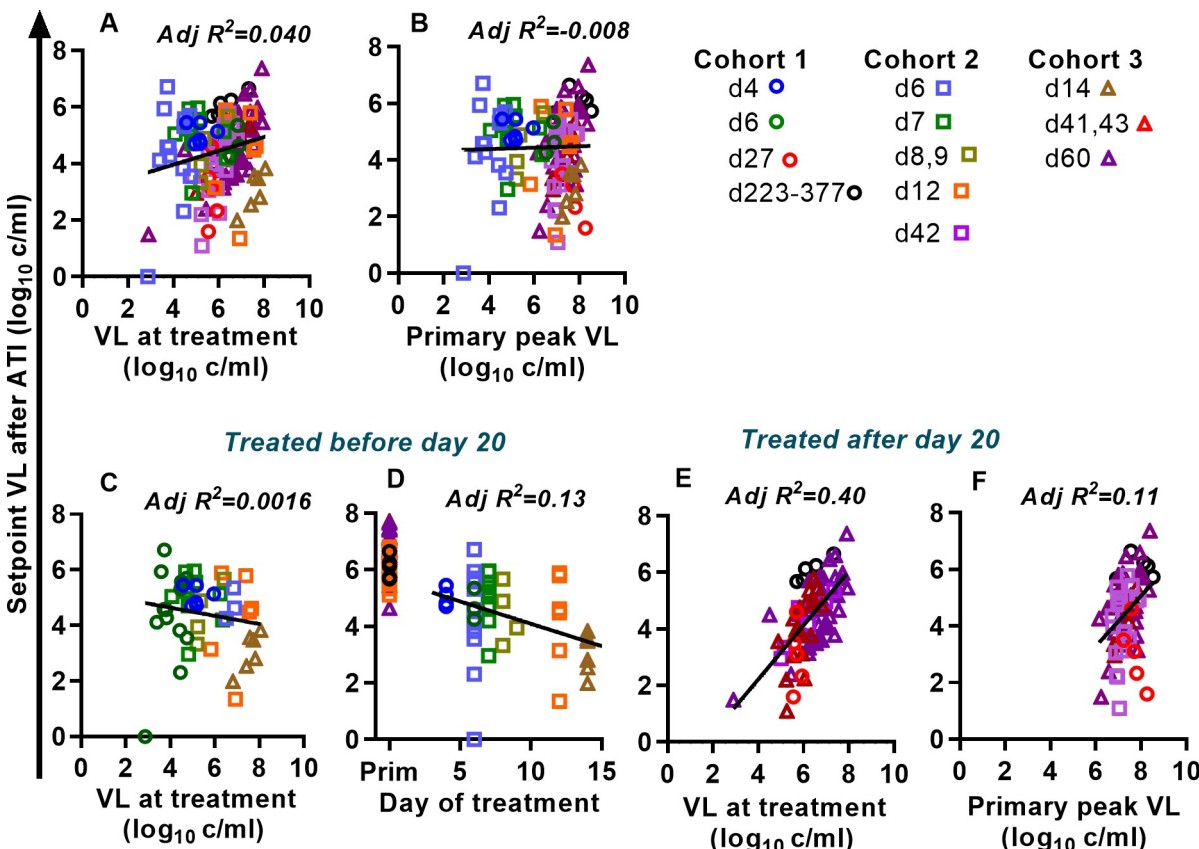

**Fig 3. Determinants of post-treatment rebound setpoint viral load.** We analysed animals from the three cohorts to understand factors associated with the post-rebound setpoint viral level. Viral load at treatment (A) and peak viral load before treatment (B) where poorly predictive of post-rebound setpoint viral load when data from all treatment times was combined. However, when we divided the animals into those treated before day 20 and after day 20, clear patterns emerged. (C,D) For animals treated before day 20, the viral load at treatment (C) does not predict setpoint viral load at rebound. Instead, the day of treatment (D) is a significant predictor of the setpoint viral load. (E,F) In animals treated after 20 days post-infection, viral load at treatment is a good predictor of the rebound setpoint viral load (E), while the rebound peak viral load is only weakly associated with rebound setpoint (F).

at treatment was a predictor of setpoint viral load during rebound (adjusted $R^2$ = 0.40, p <0.0001, see Table H in S1 Text) (Fig 3E). Interestingly, the slope of linear regression between $\log_{10}$ (viral load at treatment) and $\log_{10}$ (post rebound setpoint viral load) was very close to 1 (slope = 0.96, (95% CI 0.69–1.22)). This suggests that a doubling in viral load at treatment initiation was associated with a doubling of post rebound setpoint viral load. However, the fit also suggests that the post-rebound setpoint tends to be around 1.6 logs lower than the viral load at the time of treatment (Y-intercept of regression is -1.6 $\log_{10}$ copies/ml, CI (-3.34, 0.14)). We separately analyzed whether peak viral load in primary infection was predictive of post-rebound setpoint viral load and found that this was weakly, but significantly, associated (adjusted $R^2$ = 0.11, p = 0.0024, see Table H in S1 Text) (Fig 3F). Treatment after day 20 occurs at a time where immune responses to virus have been induced and contribute to controlling viral replication. Therefore, it seems likely that viral load at treatment (for animals initiating treatment after day 20) is indicative of overall immune competence in controlling virus, which is also contributes to the level of post-treatment control.

Although viral load at the time of ART treatment was a good predictor of post-rebound setpoint for animals treated after day 20 (Fig 3F), the timing of treatment also played a role.

Using both viral load at treatment and data on timing of treatment to the model significantly improved the fit (adjusted $R^2$ for combined model = 0.51, F-test comparing models with and without timing, p<0.0001), with increasing delay in treatment after day 20 being associated with higher post-rebound setpoint viral levels (Fig C.B in S1 Text).

## Duration of treatment is associated with increased setpoint viral levels

In addition to variation in the timing of treatment initiation, several cohorts included animals that initiated ART treatment on the same day post-infection but were treated for different lengths of time before interruption (Table A in S1 Text). This provides an opportunity to explore the effects of duration of treatment independent of the timing of treatment initiation. We used a linear mixed effects model (described by Formula (4) to investigate whether post-rebound setpoint viral load was significantly associated with the duration of treatment. We found that post-rebound setpoint viral levels increased with increasing time on therapy (fixed effect slope = 0.011, p<0.0001). The rate at which post-rebound set point viral load increased with longer time on treatment did not depend on when treatment was initiated (comparing groups treated on day 4, day 27, and days 60 (p = 0.68), see Table D in S1 Text) (Fig 4). A major caveat here is that in the Cohort 3 study, all animals were maintained with viral suppression for the same time, so there was a tendency for a longer duration of treatment to be associated with a longer 'time-to-viral suppression' on treatment. Thus, animals with a higher level of viral replication may have tended to be treated for longer, potentially confounding the analysis. Removing Cohort 3 animals from the regression analysis we obtained much weaker association of duration on treatment with setpoint viral load (fixed effect regression slope = 0.006, p = 0.1).

## Cell-associated SIV DNA, RNA, and rebound timing are poorly predictive of post-ATI setpoint viral levels

Previous studies have suggested that high levels of HIV DNA measured during ART treatment may predict subsequent poor control of viral rebound after treatment interruption [19–23].

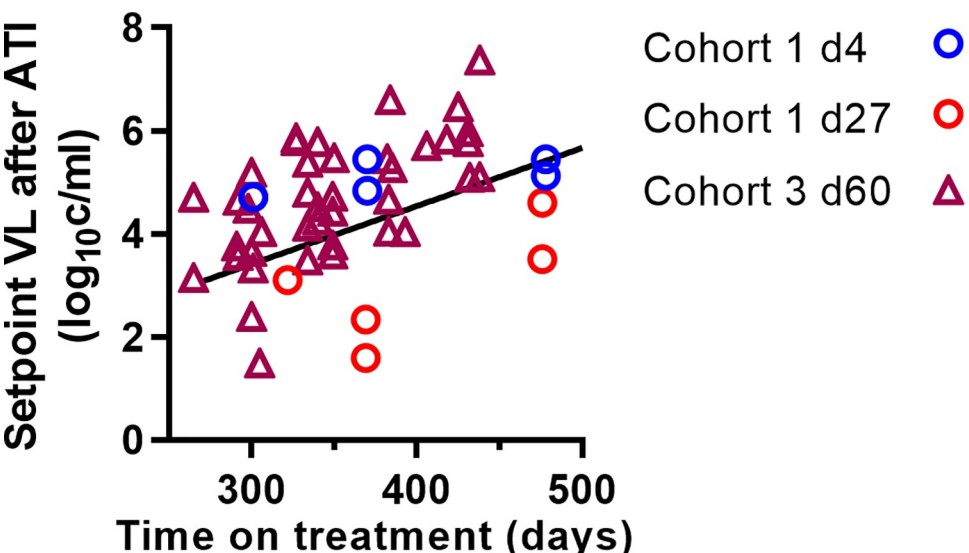

**Fig 4. Longer treatment is associated with increased post-rebound setpoint viral level.** Setpoint viral load from different cohorts treated for different amounts of time significantly correlates with time on treatment (linear mixed effect model slope = 0.011 $\log_{10}$ copies/ml per day, p<0.0001).

However, it is not clear whether a larger reservoir itself might lead to poor post-treatment control, or simply that individuals with poor control of viral replication pre-treatment tend to have higher levels of HIV DNA on treatment, and also have poor post-treatment control. To investigate this, we explored whether the level of SIV DNA in PBMC during ART just prior to treatment interruption was a predictor of the setpoint viral level after rebound. Data on pre-interruption cell-associated DNA and RNA were only available for a subset of the studies (see legend to Fig 5), all of which were treated on or before day 27. Fitting a linear mixed effect model described by Formula (4) (assuming each cohort as a random effect) to this subset of studies, we find a weak but non-significant negative correlation between the level of SIV DNA, SIV RNA just prior to ATI and post-rebound setpoint viral load (Fig 5A and 5B and Table E in

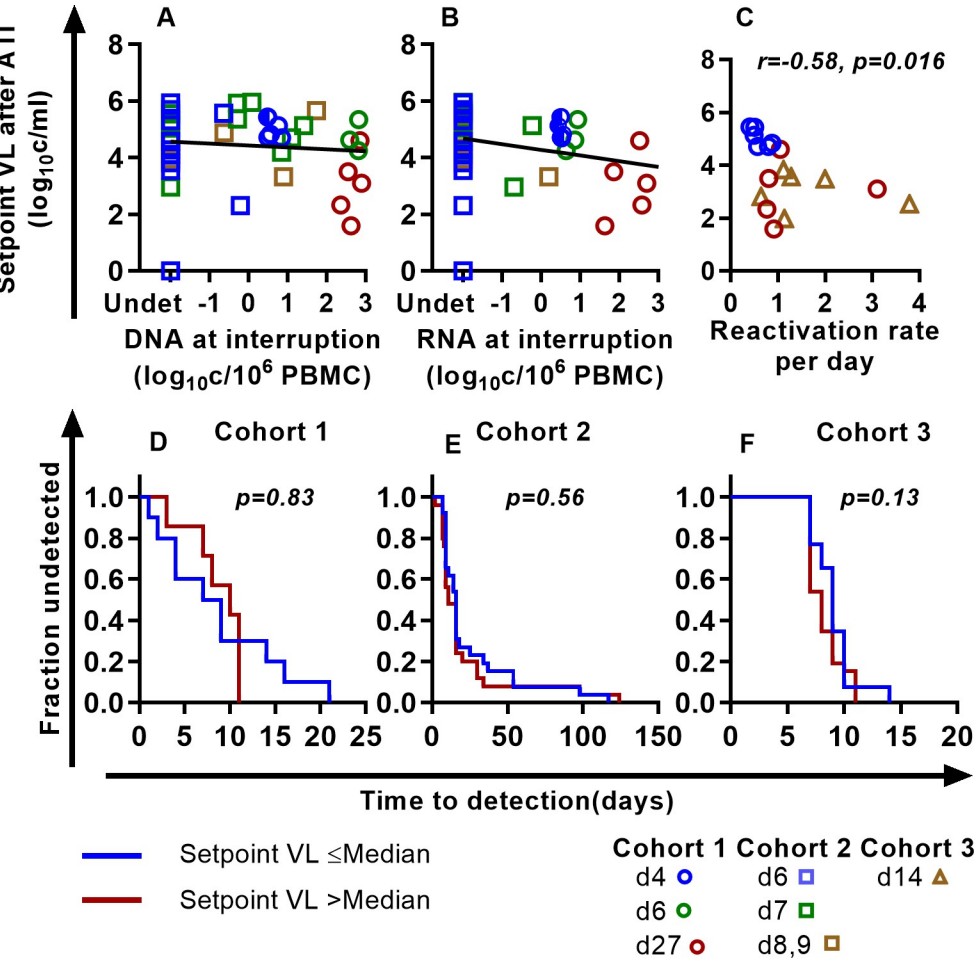

**Fig 5. Reservoir size, reactivation frequency, and early post-rebound viral setpoint:** (A, B) Weak or no association of SIV DNA (A) and SIV RNA (B) with the set point viral load at rebound. Linear mixed effect modeling (considering each cohort as a random effect for the slope and intercept of the line) finds no relationship between SIV DNA and RNA with the setpoint viral load we used. Fixed effect slopes are -0.066 and -0.202, p-values for slopes are 0.56 and 0.117 for SIV DNA and SIV RNA respectively (see Table E in S1 Text for the best-fit parameters). The frequency of rebound was directly measured in a subset of animals infected with a barcoded virus (C). Higher frequency of rebound was significantly associated with lower post-rebound setpoint viral load (r = -0.58, p = 0.016). D, E, F To investigate the relationship between time-to-rebound and post-rebound setpoint viral loads, we divided the groups into two, separating those higher or lower than the median post-rebound setpoint viral levels. Time to detection of virus after treatment interruption in monkeys with setpoint viral load less than or equal to median setpoint viral load (blue line) and greater than median (red line) is not different in any of the groups (log-rank test).

S1 Text; we also analysed cohorts separately, see Fig E in S1 Text for the results). Thus, it is clear in our meta-analysis of these studies that larger reservoir size does not directly influence high rebound viral levels.

Modelling studies have suggested that a high rate of reactivation from latency during treatment interruption may lead to poor immune control of viral rebound [19]. Since reactivation frequency affects time-to-viral rebound, we investigated whether the 'time-to-detection' of viral rebound was predictive of setpoint viral levels after treatment interruption. To do this we divided animals from each study site into those with higher than median and lower than median post-rebound SIV setpoint viral levels, and then analyzed time-to-rebound. We observed no significant difference between time-to-detection of plasma SIV in animals with higher or lower than median post-treatment setpoint viral levels, suggesting that time-to-rebound is not associated with post-rebound setpoint viral levels (Fig 5D–5F).

The frequency of individual SIV lineages rebounding from latency was also directly measured in a subset of animals infected with the barcoded SIVmac239M virus (Fig 5C). Analyzing animals treated on days 4, 14 and 27 days, we found that a higher frequency of reactivation was associated with a lower setpoint viral level (Spearman r = -0.58, p = 0.016). Again, this seems likely due to later initiation of treatment driving both higher frequencies of reactivation and higher reservoir size but also increased induction of immunity.

## CD8+ T cell responses and post-rebound viral control

The data presented above are consistent with a dynamic where delaying treatment in early infection leads to increased exposure to virus inducing higher levels of immune responsiveness, which may in turn contribute to better post-rebound control of viral replication (although as stated above, this only applies to treatment before approximately day 20 post infection). To investigate if pre-treatment immune response predicts rebound control directly, we assessed the relationship between pre-ATI SIV-specific CD8+ T cell responses and post-rebound setpoint viral loads in a subset of studies where this was available (Cohort 1 and in the Cohort 2 animals that had previously been vaccinated with a CMV-vectored vaccine (Fig 6)). The correlation between CD8 T cell responses and VL was significant in the Cohort 1 data

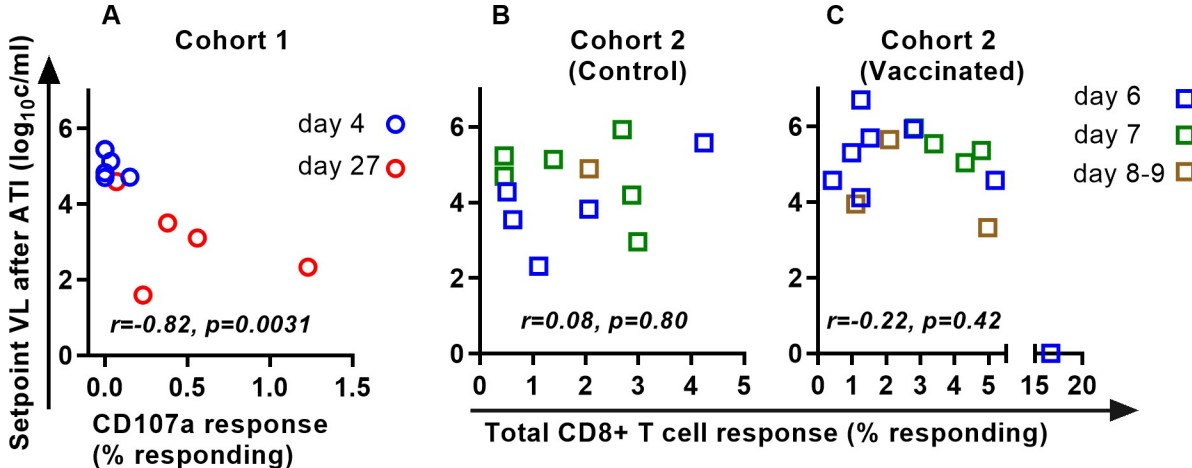

**Fig 6. Association between CD8+ T cell responses on ART and post-rebound setpoint viral load in a subset of animals in which CD8 T cell responses were measured.** A. In Cohort 1, higher CD8+ T cell responses before ATI were associated with lower post-rebound setpoint viral loads (r = -0.82, p = 0.0031). B, C. However, in Cohort 2 the frequency of SIV-specific CD8+ T cells (measured between days 511 and 609) was not significantly associated with post-rebound setpoint viral levels.

(Spearman r = -0.82, p = 0.0031). However, this seems heavily influenced by the early day 4-treated animals (low immune response and therefore poor control), and a similar trend is not obvious within the groups of animals treated slightly later. The relationship between CD8 + T cell response and viral setpoint was not significant in either the vaccinated or control animals from Cohort 2.

## Long-term control of SIV rebound

Studies in HIV have used different definitions of both the level of viral load as well as the duration of viral control to identify post-rebound 'controllers' after ATI. However, longitudinal analysis suggests that post-rebound control of viral replication in HIV is often temporary, and that the majority of 'early controllers' experience rising viral levels over time [7]. To investigate the duration of post-rebound control in our SIV model we analyzed the proportion of animals maintaining a setpoint viral level <10,000 copies/ml over time (ie: ignoring the peak of virus during the first 30 days after rebound). To account for the different lengths of follow up, we used a 'survival analysis' approach to estimate the duration of viral control (with loss of control defined as 2 consecutive viral load measurements >10,000 copies/ml) and applied censoring for animals or studies that were followed for shorter periods. We grouped animals initiating treatment around the optimal treatment time (days 14–43 post-infection) and compared these to animals treated either earlier or later in infection. Analyzing the survival curve of time until setpoint VL was >10,000 copies, the duration of control was not significantly different in animals treated day 14–43 compared to animals treated earlier (day 4–12) or later (day 60–377) (p = 0.11, log-rank test). However, analysis of the combined group as well as the individual cohorts suggested improved early viral control for animals treated between days 14 and 43 (Fig F.A-C in S1 Text). Importantly, although animals initiating treatment at days 14–43 show some evidence of early control of setpoint viral levels, this control seemed to wane with time so that by day 100 most cohorts had around 20% controllers remaining. It appears that regardless of treatment timing, around 20% of monkeys maintained long term viral control to < 10,000 copies/ml after ART interruption. Such control is rarely observed in untreated primary SIVmac239 infection.

A major challenge in treatment interruption studies in HIV is the possible pathology or transmission associated with prolonged treatment interruption. Therefore, early predictors of long-term post-rebound control might be useful to avoid the need for prolonged treatment interruption to establish setpoint viral levels. We investigated whether early viral dynamics during SIV rebound were predictive of prolonged viral control by analyzing peak viral growth rates as well as peak viral loads during rebound. Both peak viral levels and viral growth rate during rebound are good predictors of the level and duration of viral control (Fig 7B and 7C). Animals that controlled their peak viral loads during early rebound to <10,000 copies/ml also maintained good viral control over time (with over 50% maintaining <10,000 copies at 200 days). Animals with progressively higher peak viral loads showed a reduced ability to control long-term viral replication.

Animals with very low growth rates of virus (<1 day$^{-1}$) early in ATI also exhibited improved post-rebound viral control. However, for growth rates >1 day$^{-1}$, there appeared little difference between groups (with around 10–15% of animals showing long term control). This may relate to difficulties in accurately assessing growth rates in animals sampled at different times. Applying more stringent definition of control of 1000 copies/ml we observe the smaller fraction of monkeys that maintain viral load below 1000 copies/ml, however the relationship between groups remains similar to that seen with the threshold of 10,000 copies/ml (Fig F.D-F in S1 Text).

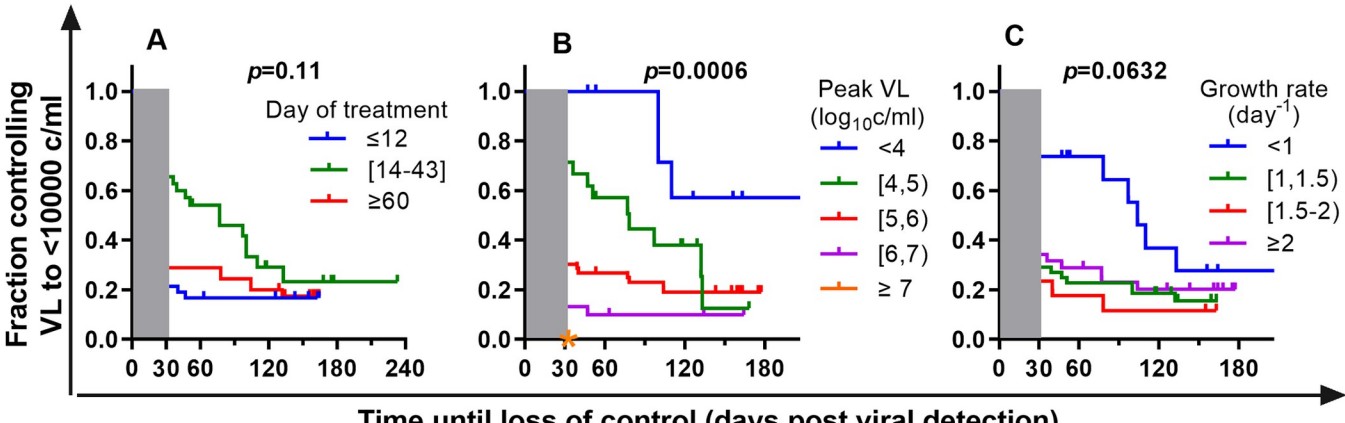

**Fig 7. Duration of post-rebound viral control to <10,000 copies/ml.** In order to compare the duration of post-rebound control, we compared the proportion of each cohort that maintained post-rebound viral loads below 10,000 copies per ml at different times after ATI. The first 30 days after detection of viral rebound are ignored to avoid the initial post-rebound peak of viral load in the analysis of the duration of viral control (shaded grey). **A.** The proportion of animals maintaining viral control over time post-rebound is higher in groups treated around the optimal time, however the difference is not significant when comparing three groups together (p-values for the log-rank test are shown in the figures). Coloured stars indicate groups where all animals had viral loads greater than 10,000 copies per ml at day 30 post detection. **B.** Animals that have a low peak of the virus during early rebound are more likely to maintain low viral control over time. **C.** Having a low viral growth rate during post-treatment rebound is also associated with longer-term control of post-rebound viral loads.

Overall, this suggests that timing of treatment had a strong effect on the level of early post-rebound control. Regardless of ART treatment timing, there appeared a 'background' level of long-term post-rebound controllers of around 20%, which occurred almost independent of the level of exposure to virus before treatment. In cohorts with 'optimal' levels of exposure, the proportion of controllers was slightly increased. However, the initial viral growth rate and viral peak during early post-treatment rebound appeared good predictors of the frequency and duration of subsequent post-rebound viral control.

## Effects of protective MHC on post-rebound viral dynamics

A number of MHC alleles including Mamu-A*01, B*08, and B*17 have been associated with lower plasma viral levels in SIV infection [24]. Animals with these alleles were scattered throughout our study (See Table A in S1 Text). To investigate the potential effects of MHC alleles on our results, we separated animals with and without these protective alleles and repeated our analysis of the relationship between time of treatment initiation and post-rebound peak and setpoint viral levels. We again compared models where we allowed different parameters to vary between the groups (model goodness of fits by AICc is summarised in Table G in S1 Text). We found that these protective alleles are associated with lower post-rebound setpoint viral levels (the setpoint viral load difference is 0.65 $\log_{10}$ copies/ml, F-test p = 0.0002) (Fig 8 and Table F in S1 Text). However, there was no significant difference in post-rebound peak viral levels (the difference is 0.067 $\log_{10}$ copies/ml, F-test p = 0.61, the best-fit parameters are in the Table F in S1 Text).

## Discussion

In this study, we investigated the dynamics of post-treatment control of viral replication in SIV infection. We observed that the setpoint viral level soon after viral rebound was strongly predicted by the timing of ART initiation during primary infection. Over the first three weeks of infection, increasing the duration of infection prior to treatment led to lower setpoint viral

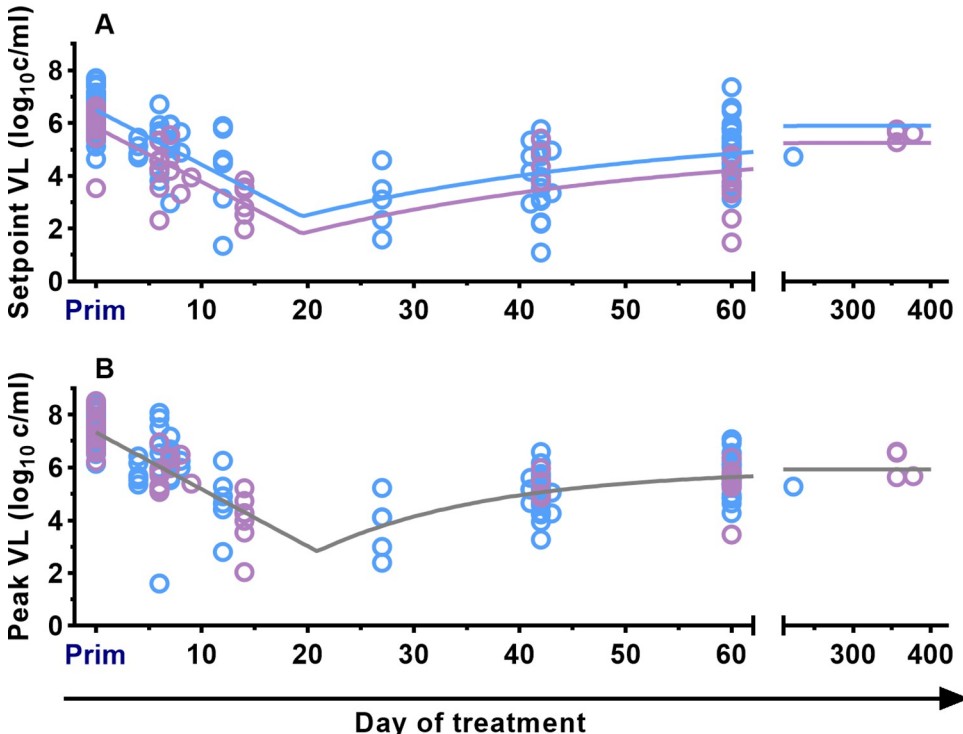

**Fig 8. Impact of protective MHC Mamu-A\*01, B\*08, and B\*17 on the setpoint and peak viral load. A.** Fitting the model defined by Formula (2) to data from macaques with any protective MHC-1 alleles A\*01, B\*08, B\*17 (purple points and curves) and macaques without these known protective alleles (blues points and curves). The curves are allowed to differ by only one parameter (the viral load at day 0 post infection–$b_0$), with the rest of the parameters fitted simultaneously to both datasets. For the data on post-rebound setpoint viral levels, the model allowing different viral levels for animals with protective alleles fits better than the model with all parameters the same. The setpoint viral load is lower in macaques with protective alleles by 0.65 $\log_{10}$ copies/ml (F-test's $p$-value = 0.0002, best-fit parameters are in Table F in S1 Text. The best-fit parameters for the model with single $b_0$ are in Table B in S1 Text). **B.** For data on post-rebound peak viral levels, the best fit model is one in which there is no difference between groups with and without protective alleles (indicated as grey curve) (F-test $p$-value = 0.61. Best-fit parameters of the model with different $b_0$ are presented in Table F in S1 Text. Best-fit parameters for the model with a single $b_0$ are in Table B in S1 Text). This suggests that the post-rebound peak viral load is not affected by the presence of protective MHC-1 Mamu-A\*01, B\*08, and B\*17 alleles.

levels after rebound, with mean viral levels post-rebound reduced around 1000-fold (compared to primary infection) in animals treated at day 14 to day 27 post-infection. However, if treatment is delayed beyond one month post-infection, post-rebound setpoint viral levels rise back towards levels seen in primary infection. This suggests a narrow window in which optimal immunity may be generated in primary infection and can lead to control of early setpoint viral load, after which immune exhaustion or viral escape may reduce the capacity to control post-rebound viral replication.

This analysis has important implications for studies of post-rebound control in SIV. Firstly, it suggests that immune interventions to control post-rebound setpoint viral levels may need to target very different mechanisms depending on whether ART was commenced early or late after infection. That is, for animals treated early in infection, our analysis suggests that

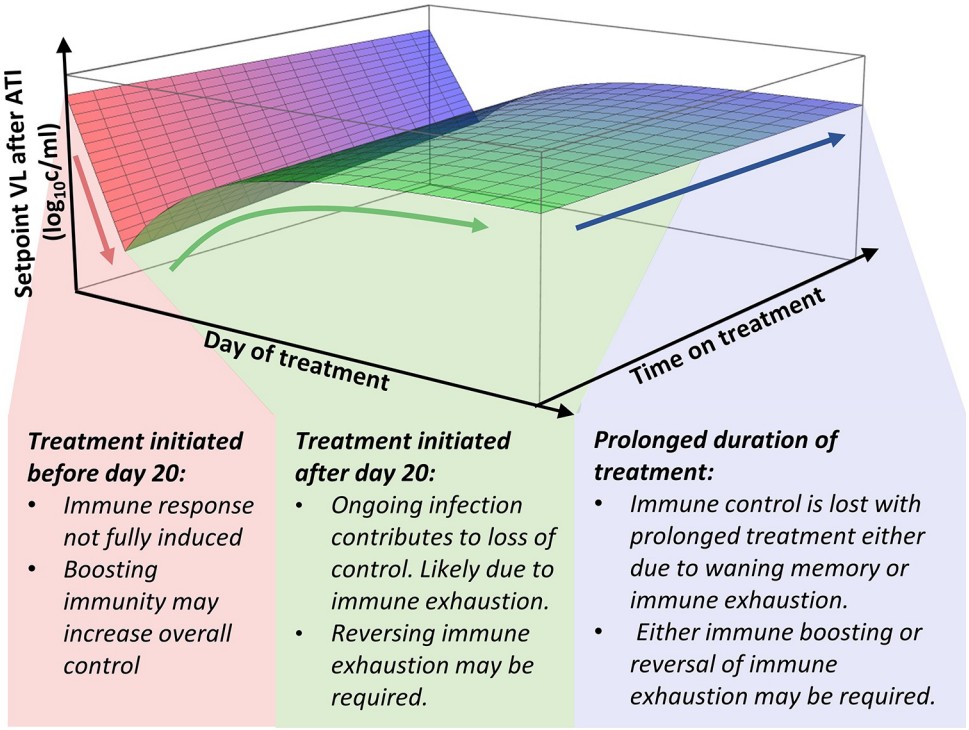

**Fig 9. Different immune interventions may be required depending on timing of ART initiation and the duration of treatment.** One approach to long-term, ART-free control of HIV is to boost immune responsiveness to infection during ART. However, different therapeutic interventions may be effective depending on the timing of ART initiation. For example, for animals treated at an early stage of infection (red shaded area), increased priming of the response or measures to boost immunity may be effective. However, for animals treated later in infection (green shaded area), it may be necessary to reverse immune exhaustion to improve post-rebound setpoint viral levels. The mechanisms that lead to increase of post-rebound setpoint after a prolonged treatment (blue shaded area) are unclear and can be explained by either declining levels of immune memory or prolonged exposure to low levels of viral antigen that drives immune exhaustion. Thus, the interventions for boosting immune control may differ, depending on the underlying mechanisms. The surface depicted here is the best-fit of Eq (3) to the data used in this study. See Fig G in S1 Text for overlay of data points on this best-fit curve.

increased exposure to virus (or viral antigen) would be expected to reduce post-rebound setpoint viral levels. Therefore, therapeutic vaccines or other interventions that can prime or boost immune response should lead to improved control of viral rebound in early treated animals. However, for animals treated later in infection it seems likely that delayed treatment and increased exposure to virus may contribute to immune exhaustion and high setpoint viral levels. Thus, in these animals, reversal of exhaustion (using checkpoint inhibitors) rather than additional exposure to viral antigens may be necessary (Fig 9). Prolonged treatment is also associated with increases in post-rebound setpoint viral levels (Fig 4), although the mechanisms of this are unclear. It may be that immune control wanes due to declining levels of immune memory. Alternatively, prolonged exposure to low levels of viral antigen may drive immune exhaustion. The ideal interventions for boosting immune control after prolonged ART may differ, depending on these underlying mechanisms.

Identifying whether the same effects of treatment timing on the dynamics of post-rebound viral levels are seen in HIV is an important future goal. Since the majority of people living with HIV are treated after peak viremia (often significantly later in infection), treatment in HIV is likely more similar to the right-hand side of the curve we observe in SIV. Thus, the major challenge for immunotherapies in HIV may be to reactivate an exhausted immune response rather

than stimulating an immune response by providing antigen directly. This contrasts with interventions in early-treated HIV, where boosting of immunity by interventions such as vaccination may be more effective. Since macaque models are an important tool in developing potential therapies, this work suggests that timing of treatment may be an important consideration in extrapolating between SIV and HIV treatment effects.

The incidence of post-rebound control and its duration are major questions in HIV infection. A recent large study has suggested that around 4% of individuals commencing ART in chronic infection show post-rebound control of HIV replication, and this may increase to as high as 13% in patients treated early in infection [7]. However, the proportion of post-rebound controllers also seems to decline over the first few years post-treatment. Our investigation of post-rebound control in macaques suggested a similar pattern to that seen in HIV (albeit over a shorter timeframe). That is, overall we observed around 30% of all macaques in our 3 cohorts showed post-rebound control of plasma viral load to below 10,000 copies ml$^{-1}$ between days 30–60 post-rebound. The proportion of animals showing control at this time was higher in animals treated in a window around the optimal treatment time (54% of animals treated between day 14–43 post-rebound showed control to <10,000 copies /ml out to day 60 (Fig 7A)). However, this higher level of early control is only temporary. By 150 days after rebound the proportion of animals with viral levels <10,000 copies / ml was similar between animals initiating treatment at different times (Fig 7A). This frequency of post-rebound controllers is higher than observed in primary SIV infection, where we did not observe viral control to <10,000 copies after day 30. Thus, the macaque model appears to recapitulate several key features of post-rebound HIV control (i) a higher proportion of controllers during rebound than primary infection, (ii) a higher proportion of early post-rebound controllers following early treatment, (iii) peak viral levels during rebound are predictive of both level and duration of viral control, and (iv) there is a general loss of control of viral replication over time.

Measurement of viral growth rates during rebound was also associated with post-rebound control, with 28% of animals with growth rates <1 day$^{-1}$ showing sustained post-rebound control, compared to 17% of animals with higher growth rates. However, accurate measurement of HIV growth rates during rebound requires frequent measurement, and there is as yet not enough data to determine whether rebound growth rate predicts setpoint viral loads in HIV. Importantly, it is also not clear whether lower viral growth rates directly contribute to post-rebound control (i.e., slow viral growth facilitates immune control), or are simply reflective of better existing immune control. This is important in considering interventions such as bNAb therapy, that may act to slow viral growth during rebound. If reducing viral growth *per se* improves the development of immunity and subsequent viral control, then there may be a benefit to slowing rebound growth rate. However, if this is not causative, then slowing early rebound viral growth may have little long-term effect.

Our study aggregates data from 10 previous studies from three independent research institutions. This allows us to consider patterns of latency and rebound across a larger cohort than would be possible in a single study. However, it also requires combining data from studies that differ in known factors such as challenge dose, drug treatment schedule, as well as potential unknown factors such as macaque microbiome. However, the combined dataset also allows us to confirm that patterns are seen consistently across three independent sites. It is important to consider potential confounding factors in the study. For example, one factor potentially affecting SIV rebound viral levels is the macaque MHC, and we show that the protective MHC are associated with a lower post-rebound setpoint viral level. Importantly, the same relationship between time of treatment initiation and post rebound peak and setpoint viral levels is observed independently in animals grouped by MHC type, reinforcing that the result is

independent of this effect. Another factor is that some monkeys were infected intrarectally, and mucosal infection may affect early viral kinetics compared with IV infection. However, the only group with mucosal infection initiated treatment on days 223–377 (Table A in S1 Text), and it seems unlikely that this minor differences in early viral kinetics would affect our conclusion. Despite the limitations of this study, it also provides a framework for investigating the mechanisms of immune control of viral rebound. For example, identifying the differences in the magnitude, phenotype, and anatomical location of immune responses for animals treated at different times post-infection may identify the primary mechanisms of post-treatment control. Further studies should aim to confirm these findings and investigate the mechanisms of post-treatment control of SIV replication in a large and homogeneous cohort of animals.

Macaque models of HIV infection, treatment, latency, and reactivation provide a powerful tool to investigate both the natural history and pathogenesis of infection, as well as to investigate the effects of interventions. In our previous studies we have investigated how timing of treatment can affect the frequency of rebound from latency [17]. Here we extend that work to understand how factors such as timing and duration of treatment can affect post rebound control of viral replication. If a similar association between timing of treatment initiation and post-treatment control of viral replication can be established for HIV, this may have major implications for future treatment strategies.

## Material and methods

### Ethics statement

This study includes previously published and unpublished data from studies of SIV infection of 124 rhesus macaques. Data was aggregated from 10 independent studies across 3 institutions. Protocols were approved by the relevant institutional committees: the Institutional Animal Care and Use Committee of the National Cancer Institute, National Institutes of Health (NIH) in NIH-Bethesda facilities, the Oregon National Primate Research Center's Animal Care and Use Committee, or Emory University's Institutional Animal Care and Use Committee (as detailed below and in Table A in S1 Text).

### Animal models of SIV infection

The details of all cohorts can be found in the Table A in S1 Text.

**Cohort 1.** This cohort incorporated data from 19 macaques from a study conducted by the Frederick National Laboratory. Fourteen animals were infected intravenously with $2.2 \times 10^5$ IU of barcoded SIVmac239M and subsequently treated with combinations of tenofovir (TFV), emtricitabine (FTC), raltegravir (RAL), indinavir (IDV), and ritonavir (RTV), or of TFV, FTC, and RAL, starting between days 4 and 27 (studies 1.a-1.c) for 301 to 478 days. Details of the studies that involved these animals were previously published [16]. The five remaining animals (study 1.d) were part of a study that has not been previously reported. These animals were infected intrarectally with $3.0 \times 10^2$–$1.0 \times 10^4$ of the genetically tagged SIVmac239X virus and subsequently treated with a combination of tenofovir disoproxil fumarate (TDF), FTC, dolutegravir (DTG), and darunavir (DRV) following a period of DRV monotherapy. Suppressive combination therapy started between days 223 to 377 and was maintained for 218 to 410 days (see Table A in S1 Text for details [16,17]). For these studies, all work involving research animals was conducted under a protocol approved by the Institutional Animal Care and Use Committee of the National Cancer Institute, National Institutes of Health (NIH) in NIH--Bethesda facilities. NIH-Bethesda is accredited by AAALAC International and follows the Public Health Service Policy for the Care and Use of Laboratory Animals (Animal Welfare

Assurance Number D16-00602). Animal care adhered to the standards outlined in the "Guide for the Care and Use of Laboratory Animals (National Research Council; 2011; National Academies Press; Washington, D.C.), in accordance with the Animal Welfare Act.

**Cohort 2.** This cohort incorporated data from 57 rhesus macaques from The Oregon National Primate Research Center, as previously reported [11]. Studies were conducted with approval from the Institutional Animal Care and Use Committee, under the standards of the US National Institutes of Health Guide for the Care and Use of Laboratory Animals. Animals were infected with two TZM-bl assay focus-forming units of SIVmac239X and treated with combinations of TDF, FTC, DTG or TDF, FTC, DTG, DRV, RTV, beginning between days 6 and 42 for 606–923 days (studies 2.a-2c). Animals in day 6–9 and 42 treated groups (study 2.a) were also vaccinated with CMV/SIV (n = 27) or control RhCMV vectors (n = 18). Both subgroups have no statistically significant difference in the parameters of interest (see Fig H in S1 Text for comparison).

**Cohort 3.** This cohort incorporated data from 53 macaques from Emory National Primate Research Center (EPC). All studies were reviewed and approved by Emory University's Institutional Animal Care and Use Committee (IACUC) under permit numbers 201700655, 201700007 [18], 3000065, 2003297, 2003470, and 201700665 [15]. Animal care facilities at the Emory National Primate Research Center (EPC) are accredited by the U.S. Department of Agriculture (USDA) and the Association for Assessment and Accreditation of Laboratory Animal Care (AAALAC) International. Seven animals (study 3.a) were infected with $1 \times 10^4$ IU of barcoded SIVmac239M and treated on day 14 for 205 or 206 days with the combination of FTC, TDF (tenofovir disoproxil fumarate), DTG. Six animals were infected with 300 TCID50 of SIVmac239 and treated on days 41 or 43 for duration of 357 days with combination of FTC, TDF, DTG (study 3.b) [18]. Forty-one macaques were infected with 300 TCID50 of SIVmac239 and treated with the combination therapy of FTC, TDF, DTG on day 60 post-infection for duration 265–438 days [15] (study 3.c). Thirty-five monkeys were also treated with immune checkpoint blockade (ICB) which did not affect the parameters discussed in this study (see Fig H in S1 Text for comparison of parameters in treated and control groups).

## Mathematical analysis

### Estimation of the setpoint

We defined primary infection and post-rebound $\log_{10}$ setpoint viral load as the time-weighted area under the curve of $\log_{10}$ viral load between 30 and 60 days after the first detection of viral rebound in the blood (in order to avoid the initial peak of rebound viral levels and allow comparison in an interval that is available in most animals in our study). Where animals with follow up shorter than 60 days post rebound are included in the analysis, this is stated in the relevant text of the manuscript and in the Table A in S1 Text. The average viral load at the setpoint ($V_s$) is calculated according to the Formula (1)

$$\log_{10} V_s = (\int_{t_s}^{t_e} \log_{10} V(t) \mathrm{d}t) / (t_e - t_s) \tag{1}$$

Where $t_s$ is the time of the start of setpoint (30 days after detection of viral rebound) and $t_e$ the time of the end of setpoint (60 days). $V(t)$ is the viral load at time $t$. Since the viral load is not continuously measured in the experiments, $V(t)$ is estimated by linear interpolation of $\log_{10}$-transformed viral load data.

## Estimation of the duration of viral control

We defined the duration of control as a length of the time interval between the initial 30 days post-detection and the first measurement where viral load exceeds the control threshold of $10^4$ copies/ml for at least two consecutive measurements. The details of estimation of the duration of control are in the Supplementary method in S1 Text.

## Linear regression

Liner regression and F-test of one and two-variable models were performed in Wolfram Mathematica 11.2 (Wolfram Research Inc, Champaign, IL, USA) using standard function LinearModelFit.

## Non-linear regression to determine how treatment timing affects viral kinetics

In order to find the inflection point of the $\log_{10}$ setpoint and peak viral loads, we proposed a nonlinear piece-wise function. Firstly, the value of function $g(t)$ drops linearly with the parameter $b_0$ and $b_1$ until some minimal level attained at the timepoint $t_{min}$, and then it grows with the rate $k_1$ before plateauing at some maximal level $b_0 + \delta$ as described by the Formula (2).

$$g(t) = \begin{cases} f_1(t) = b_0 + b_1 t, t < t_{min}, \\ f_2(x) = (1 - \exp(-k_1(t - t_{min}))(b_0 + \delta - f_1(t_{min})) + f_1(t_{min}), t \geq t_{min}. \end{cases} \quad (2)$$

We fitted Formula (2) to the $\log_{10}$-transformed peak and set point viral loads assuming that setpoint at primary infection corresponds to the point when the day of treatment is equal to 0. The fitting and F-test was performed using GraphPad Prism v 9.4.

Peak and set point viral loads after rebound were also fitted with Formula (2) using different maximal values ($b_0$) for animals with or without a protective MHC-1 allele (A*01, B*08, and/ or B*17).

For the purposes of visualisation, we also fitted an extended model that included an increase in post-rebound setpoint viral load with time-on-treatment. To do this, we modified function (2) by adding the effect of duration of treatment ($\tau$) as shown by Formula (3)

$$h(t, \tau) = (b_0 + \delta - g(t))(1 - \exp(-k_2 \tau)) + g(t), \quad (3)$$

where $k_2$ is the rate of the decline of protection on treatment. Formula (3) was fitted to the data using Wolfram Mathematica 12.3, Wolfram Research Inc, Champaign, IL, USA using standard function NonlinearModelFit.

## Linear mixed effects model

In order to determine the existence of linear relationships between variables of interest in different groups and whether these relationships are statistically significant significantly (slope$\neq$ 0), we used a linear mixed effects model of the form.

$$y_{ij} = a + a_i + (b + b_i)x + \varepsilon_{ij} \quad (4)$$

Where $a$ and $b$ are fixed effects of intercept and slope and $a_i$ and $b_i$ are random effects of each group of monkeys (indexed by $i$), and $\varepsilon_{ij}$ is the normally distributed variation. The regression was implemented in R using the standard function lme() from library nlme. In order to test if slopes are significantly different (i.e. $b_i \neq 0$ vs. $b_i = 0$) or whether the model with random

effects fits better than the model with only fixed effects we implemented the standard R function—anova().

## Statistical tests

Coefficients of correlation as well as all statistical tests except goodness of fit test for regressions mentioned above were performed in GraphPad Prism v 9.4.

## Supporting information

**S1 Text. Supplementary text.** Summary of the experimental studies and additional analysis. (PDF)

## Acknowledgments

For NHP studies conducted at ENPRC, we gratefully acknowledge Chris Parry at ViiV Healthcare for supplying DTG, and Romas Geleziunas at Gilead Sciences for supplying FTC and TDF. For cohort 1, study 1.d, conducted at NCI, DTG was kindly provided by Viiv Healthcare. Non-barcoded SIVmac239 viral stocks used in cohort 3 (studies 3.b and 3.c) were provided by Koen Van Rompay at UC-Davis. Viral load measurements for the entire study were performed by the Quantitative Molecular Diagnostics Core at Frederick National Laboratory and at the Emory Center for AIDS Research.

## Author Contributions

**Conceptualization:** Mykola Pinkevych, Miles P. Davenport.

**Data curation:** Steffen S. Docken, Christine M. Fennessey, Gregory Q. Del Prete, Maria Pino, Justin L. Harper.

**Formal analysis:** Mykola Pinkevych, Miles P. Davenport.

**Funding acquisition:** Michael R. Betts, Mirko Paiardini, Brandon F. Keele, Miles P. Davenport.

**Investigation:** Afam A. Okoye, Christine M. Fennessey, Gregory Q. Del Prete, Maria Pino, Justin L. Harper, Michael R. Betts, Mirko Paiardini.

**Methodology:** Mykola Pinkevych, Miles P. Davenport.

**Project administration:** Steffen S. Docken, Miles P. Davenport.

**Software:** Mykola Pinkevych.

**Supervision:** Miles P. Davenport.

**Writing – original draft:** Mykola Pinkevych, Steffen S. Docken, Miles P. Davenport.

**Writing – review & editing:** Mykola Pinkevych, Steffen S. Docken, Afam A. Okoye, Christine M. Fennessey, Gregory Q. Del Prete, Maria Pino, Justin L. Harper, Michael R. Betts, Mirko Paiardini, Brandon F. Keele, Miles P. Davenport.

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
