## [Decision Letter · Decision Letter 0]

16 Apr 2023

Dear Prof. Davenport,

Thank you very much for submitting your manuscript "Timing of anti-retroviral therapy predicts post-treatment control of SIV replication" for consideration at PLOS Pathogens. As with all papers reviewed by the journal, your manuscript was reviewed by members of the editorial board and by several independent reviewers. In light of the reviews (below this email), we would like to invite the resubmission of a significantly-revised version that takes into account the reviewers' comments.

We cannot make any decision about publication until we have seen the revised manuscript and your response to the reviewers' comments. Your revised manuscript is also likely to be sent to reviewers for further evaluation.

Sincerely,

Robert F. Siliciano

Guest Editor

PLOS Pathogens

Susan Ross

Section Editor

PLOS Pathogens

Kasturi Haldar

Editor-in-Chief

PLOS Pathogens

orcid.org/0000-0001-5065-158X

Michael Malim

Editor-in-Chief

PLOS Pathogens

orcid.org/0000-0002-7699-2064

Reviewer's Responses to Questions

**Part I - Summary**

Reviewer #1: Pinkevych and coworkers have addressed the issue of identifying determinants and predictors of SIV control after interruption of anti-retroviral therapy. These questions are important ones for understanding how viral control off therapy can be enhanced and sustained. They pooled data from different institutions and 10 different animal cohorts that comprised 124 SIV-infected/ ART-treated rhesus macaques that underwent a subsequent ATI. All animals were infected with SIVmac239 (± barcodes), although there were differences in the inoculating dose, route of infection, duration of treatment, and experimental designs of the pooled studies. Nonetheless, from this meta analysis, themes emerged in that 1) when treatment was given before 3 weeks of infection, the longer the time to initiate treatment, the lower the post ATI peak and set point; and 2) when treatment was started after 3 weeks of infection, the longer time to initiate treatment correlated with higher post-ATI set points. They argue that these different outcomes, dependent on the time of ART initiation, likely result from a maturation of immune responses during the first 3 weeks (i.e. the longer the better for post ATI-control) whereas after 3 weeks, these immune responses are not operative, likely due to immune exhaustion (i.e. the longer the time for starting therapy, the worse for post ATI-control).

As the authors note, NHP models offer advantages for addressing viral dynamics post ATI, and set points are rarely established given the medical need and ethics for ART to be re-initiated in humans.

It is a challenging undertaking to bring together data from multiple NHP studies from diverse experimental protocols, designed to address different questions. Although the trends the authors have identified are intriguing, and their hypothesis that early functional and later dysfunctional immune responses underlie their findings, there are alternative explanations and aspects that are not easily controlled for in this analysis of pooled data. These confounding issues are noted throughout the manuscript, and include the presence of controlling MHC alleles in animal cohorts, the timing and nature of ART used, and the extent to which viral escape is occurring.

Reviewer #2: This work is very interesting and highly relevant regarding reservoir size before treatment initiation and viral load rebound after treatment interruption. The authors fit a simple model to viral load data of three different cohorts to analyze rebound post-treatment interruption and the association between pre-treatment peak viral load, the timing of treatment, and viral set point post-treatment interruption. The authors found time-dependent post-treatment control and set point viral load, where the viral load was better controlled when treatment was initiated within the first three weeks post-infection. Treatment initiation after three weeks led to higher post-rebound set point viral levels. Furthermore, the authors state that the peak viral load at treatment initiation is a primary predictor for post-treatment control and set point viral load.

I have some comments and suggestions regarding the manuscript.

The part about data explanation is excellent. However, the part about model analysis takes much work to follow. The modeling part of the manuscript is not reproducible in its current form. Furthermore, most model interpretations are based on weak associations, and the different results should be put into context with each other. The different results about treatment timing are very confusing. Please see the following comments.

**Part II – Major Issues: Key Experiments Required for Acceptance**

Reviewer #1: 1. Beyond the data in Figure 1 showing the relationship of viral set point (VSP) and peak after ATI and time of ART initiation, many of the points discussed in groups and subgroups are complicated, and correlations or lack of correlations are difficult to tease out. For example, set point after rebound is later divided into early set point (reduced with early Rx) vs later set point. It seems that a Summary Table would be helpful with categories of

Time to viral rebound

VSP post rebound

Peak post rebound

Control post rebound

vs. correlates or non-correlates listed for

Early initiation of Rx

Late initiation of Rx

Duration of ART

Viral Peak during primary infection

Cell-associated DNA and RNA

CD8 T cell responses

2. The issue of controlling MHC alleles, which could be a major confounder in this analysis, was, surprisingly, not mentioned in the Results section. While this was discussed in the Discussion with data in a Supplemental Figure, it should be presented in Results and with data shown as a Figure. This reviewer still has concerns that effects of controlling or non-controlling alleles could be contributing to some of the findings that are discussed.

3. The reservoir/cure field intuits that immune control is operative during ATI and shapes the dynamics of rebound. The authors argue that these findings indicate that approaches to post treatment control should be different for individuals treated early vs treated late. Given that HIV infected individuals rarely are treated within 3 weeks of infection, the main message for humans is that immune dysregulation, exhaustion, viral escape are likely the drivers of relapse post ATI. What new information is being added here that is relevant for next steps in humans?

Reviewer #2: Fig 1A: Even though differences between the cohorts are mentioned in the Methods section, it would be helpful to see all viral load trajectories as a supplementary figure, not only selected of cohort 1. Especially study 1d, which has yet to be published, should be illustrated in the supplement (besides all other studies and cohorts). Regarding the cohort-specific results, there appears to be cohort-specific differences. Especially if you pick different cohorts and studies for different analysis. Also, please refer to Fig 1A in the text and explain the data.

Lines 104 and 105: “shorter periods were used in some animals due to data availability”: This is not clear regarding reproducibility. What is the analyzed time frame? 30 to 60 days or animal-specific? You refer to Table S1, but where is the information on the length of the analyzed time frame?

Lines 121 and 129: You state in line 121 that “greatest control is estimated to occur around 18.5 days post-infection”. I do not understand the statement in line 129 that the viral set point is increased if treatment is delayed beyond day 18 post-infection. Both statements seem conflicting. Please explain a little more.

Lines 149+: You mentioned that the peak viral load was 0.98 log10 copies per ml higher than set point, regardless of treatment initiation. What does that imply? Can you explain what that means? Additionally, the colors in Supplementary Figure 2 are hard to discriminate. Please use different colors. It is also unclear what models were tested, the goodness of fit, AIC or BICc, and how the parameters differed. Please report those values or values of model comparison. The data looks noisy, and more than one model may fit the data.

Line 174: As mentioned earlier, a supplementary figure showing the viral trajectories of Cohort 2b of all cohorts might be helpful.

Lines 182 to 184: Why split data on day 20? Is that related to the 18.5 days mentioned in line 121 or the 20.9 days in line 145? Did you also try other splits, such as day 7?

Lines 191 and 192: You mention a significant association between the timing of treatment and post-rebound set point viral levels. Even though statistically significant, the association of R2 = 0.13 is weak and should be mentioned. You refer to that in lines 201 to 203 that timing of treatment is a major predictor of post-rebound set point viral levels in animals treated before day 20 post-infection. However, this whole analysis could be more convincing, considering that R-square values show relatively weak associations.

Line 207: You mentioned that R2 = 0.40 is a strong predictor. An R2 = 0.4 is not strong but rather low to average.

Line 239 and Fig 3: Why did you only analyze cohort 1 d4, d27 and cohort 3 d60? Why did you not use all the data? You can also use segmented regression combined with a mixed-effect model if linear regression for all data does not work well. Also, what are the group-specific slopes (variability)?

Fig 4A to 4D: Again, there seem to be differences between the cohorts affecting the analysis. As mentioned above, seeing the viral load trajectories of all cohorts and treatment groups would be helpful.

Line 317 to 319: “animals treated around the optimal time”? Are you referring to the 18.5 days you mentioned earlier? What is the optimal time?

Fig 6A, B, and C: Do you have any suggestions as to why animals from cohort 1 treated on d4, d6, and d223-377 are poor controllers? Can you give more explanations about what is going on in the figure?

Fig 6D: You do not mention Fig 6D. Please explain Fig 6D.

Fig 6E and 6F: Since you split the data earlier in treatment <= 20 days and > 20 days, you could also perform the “survival analysis” for both treatment groups.

Lines 397 to 400: “overall we observed around 30% of macaques showed early post-rebound control”. Where is that number coming from? You talked about 20% controllers, which seems to be a background level of long-term control. Please explain where the 30% is coming from. Moreover, “the proportion of early controllers was higher in animals treated in a window between day 15 - 41 post-rebound”? I do not understand that sentence. Is days of treatment 15 to 60? And also seems to conflict with the result earlier in line 121 that “greatest control is estimated to occur around 18.5 days post-infection” (or between days 14 and 27 in line 114). Can you explain how this all fits together?

Supplementary Figure 5: It is interesting that different models work for A and B, where there is a difference between the MHC alleles in set point VL (A) but no difference in peak VL (B). The data seems equally distributed. Visually, there also appears to be no difference in A. Did you test models that differ in multiple parameters (not only b0)? Please report goodness of fit, AIC, or BICc for model comparison (or other values of model comparison) for both set point and peak VL and the models you tested. Additionally, it is tough to gather the parameter information since they are in different tables and mixed with different results. Please make it easier for the reader to understand your results.

**Part III – Minor Issues: Editorial and Data Presentation Modifications**

Reviewer #1: 1. This statement in the legend of Figure 1 is not clear: “The duration of treatment for the setpoint and peak viral load at primary infection (Prim) is assumed to equals 0 days.”

2. It should be noted that for animals mucosally inoculated, the time for actual infection could be less clear than indicated in Figure 1.

3. Line 688: “Vertical dashed lines indicate days of treatment” should be stated, “… days of treatment initiation.”

Reviewer #2: - refer to Fig 1A

- In line 77 you mention the “Fiebig 1” stage: Not all readers, especially those outside HIV, may be familiar with the Fiebig stages. Please add one sentence for the Fiebig I stage or add a reference

- line 316: copies per ml

- Line 517 and 518: Specify the limit of detection

PLOS authors have the option to publish the peer review history of their article (what does this mean?). If published, this will include your full peer review and any attached files.

Reviewer #1: No

Reviewer #2: No
---

## [Decision Letter · Decision Letter 1]

1 Aug 2023

Dear Prof. Davenport,

Thank you very much for submitting your revised manuscript "Timing of initiation of anti-retroviral therapy predicts post-treatment control of SIV replication" for consideration at PLOS Pathogens. Your manuscript was reviewed by the previous reviewers and both felt that you had addressed their concerns. One reviewer asked for some minor revisions to the manuscript that should be addressed. Based on the reviews, we are likely to accept this manuscript for publication, providing that you modify the manuscript according to the review recommendations.

Sincerely,

Susan R. Ross, PhD

Section Editor

PLOS Pathogens

Susan Ross

Section Editor

PLOS Pathogens

Kasturi Haldar

Editor-in-Chief

PLOS Pathogens

orcid.org/0000-0001-5065-158X

Michael Malim

Editor-in-Chief

PLOS Pathogens

orcid.org/0000-0002-7699-2064

Reviewer Comments (if any, and for reference):

Reviewer's Responses to Questions

**Part I - Summary**

Reviewer #1: Pinkevych and coworkers have provided detailed and helpful responses to the criticisms and suggestions of both reviewers, resulting in a much improved manuscript. Their study of SIV-infected and ART-suppressed macaques is novel in analyzing data from 10 different animal cohorts to address correlates of control following discontinuation of antiretroviral therapy. As they note, NHP models offer advantages for addressing viral dynamics post ATI in that set points following rebound cannot be established in humans given medical and ethical issues for ART to be reinitiated. The key points of the paper are well stated, namely that 1) when treatment was given before 3 weeks of infection, the longer the time to initiate treatment, the lower the post ATI peak and set point; and 2) when treatment was started after 3 weeks of infection, the longer time to initiate treatment correlated with higher post-ATI set points. These interesting patterns seen across multiple cohorts suggest different underlying mechanisms for viral control following early vs. late treatment, and reasonable hypotheses are proposed. The impact of MHC alleles on these findings is also started more clearly, and it is impressive that these same patterns were seen even when controlling alleles were grouped. There are also many somewhat surprising findings including that 1) plasma viral loads prior to treatment were not predictive of post-rebound kinetics; and 2) larger reservoir size did not directly influence high rebound viral levels.

Overall, the manuscript has been improved in clarity and in driving home the importance of these findings in considering mechanisms that could be relevant for control of HIV post-ATI.

I have only minor issues for them to address.

Reviewer #2: The authors have revised the manuscript and followed my suggestions. I appreciate your work and I recommend this manuscript to be published in this journal.

**Part II – Major Issues: Key Experiments Required for Acceptance**

Reviewer #1: (No Response)

Reviewer #2: (No Response)

**Part III – Minor Issues: Editorial and Data Presentation Modifications**

Reviewer #1: 1. I did found it helpful to have the new summary table listing correlations (or lack thereof) as Supplemental Table 8. This provides an anchor for the reader to take the breadth of diverse findings that were presented throughout the Results. However, there is data presented in Results Section that does not refer to Table 8. In fact, Suppl. Table 8 is only cited once in the manuscript (Line 221). The authors should cite Suppl Table 8 wherever appropriate, when data summarized in this Table are described in the Results Section.

For example, Line 247 (“viral load at treatment was a predictor of setpoint viral load during rebound (adjusted R2 = 0.40, p <0.0001) (Figure 3E) and Line 255 (“ … found that this was weakly, but significantly, associated (adjusted R2 = 0.11, p=0.0024) (Figure 3F”) should refer to Suppl. Table 8.

There are also points made in Results that should be added to this Summary Table: Line 763 Figure 5- “… no relationship between SIV DNA and RNA with the setpoint viral load we used. Fixed effect slopes are -0.066 and -0.2015763, p-values for slopes are 0.56 and 0.117 for SIV” ; and Line 315- “We observed no significant difference between time-to-detection of plasma SIV in animals with higher or lower than median post-treatment setpoint viral levels, suggesting that time-to-rebound is not associated with post-rebound setpoint viral levels (Figure 5D-F)”

Also, for clarity, the language of this Table should fit exactly the language in the text (i.e. change “viral load at treatment” in Suppl Table 8 to “viral load at treatment initiation”; change “primary peak viral load” in the Table to “peak viral load in primary infection.”)

2. The authors use a rather generous definition of post rebound "control" with a set point of <10,000 copies/ml, seen in 20% of monkeys maintained on long term viral control. They note that this is rarely observed in untreated primary SIVmac239 infection. Were there any animals with more impressive levels of control (e.g. <1,000 copies/ml)? If so, it would be interesting to note if there was anything special about such animals even if the numbers (of animals) did not rise to a level of statistical significance.

3. In the Discussion, Line 513 states: “In our previous studies we have investigated how

timing of treatment can affect the frequency of rebound from latency.” Please list the reference to which you are referring.

Reviewer #2: (No Response)

PLOS authors have the option to publish the peer review history of their article (what does this mean?). If published, this will include your full peer review and any attached files.

Reviewer #1: No

Reviewer #2: No

Figure Files:

Data Requirements:

Reproducibility:

References:

---

## [Editor Report · Decision Letter 2]

4 Sep 2023

Dear Prof. Davenport,

We are pleased to inform you that your manuscript 'Timing of initiation of anti-retroviral therapy predicts post-treatment control of SIV replication' has been provisionally accepted for publication in PLOS Pathogens.

Best regards,

Robert F. Siliciano

Academic Editor

PLOS Pathogens

Susan Ross

Section Editor

PLOS Pathogens

Kasturi Haldar

Editor-in-Chief

PLOS Pathogens

orcid.org/0000-0001-5065-158X

Michael Malim

Editor-in-Chief

PLOS Pathogens

orcid.org/0000-0002-7699-2064
---

## [Editor Report · Acceptance letter]

22 Sep 2023

Dear Prof. Davenport,

We are delighted to inform you that your manuscript, "Timing of initiation of anti-retroviral therapy predicts post-treatment control of SIV replication," has been formally accepted for publication in PLOS Pathogens.

Best regards,

Kasturi Haldar

Editor-in-Chief

PLOS Pathogens

orcid.org/0000-0001-5065-158X

Michael Malim

Editor-in-Chief

PLOS Pathogens

orcid.org/0000-0002-7699-2064